# Exosomes and miRNAs in Cardiovascular Diseases and Transcatheter Pulmonary Valve Replacement: Advancements, Gaps and Perspectives

**DOI:** 10.3390/ijms252413686

**Published:** 2024-12-21

**Authors:** Runzhang Liang, Naijimuding Abudurexiti, Jiaxiong Wu, Jing Ling, Zirui Peng, Haiyun Yuan, Shusheng Wen

**Affiliations:** 1Department of Cardiovascular Surgery, Guangdong Provincial People’s Hospital (Guangdong Academy of Medical Sciences), Southern Medical University, Guangzhou 510080, China; alz2901717@163.com (R.L.); jiaxiongw@outlook.com (J.W.); 2Department of Cardiovascular Surgery, Guangdong Cardiovascular Institute, Guangdong Provincial People’s Hospital, Guangdong Academy of Medical Sciences, Guangzhou 510080, China; njmd4224@163.com (N.A.); lingjing@gdph.org.cn (J.L.); pzrdoct@163.com (Z.P.)

**Keywords:** extracellular vesicle, exosome, miRNA, transcatheter pulmonary valve replacement, cardiovascular diseases, pulmonary regurgitation

## Abstract

As an important carrier of intercellular information transmission, exosomes regulate the physiological and pathological state of local or distant cells by carrying a variety of signal molecules such as microRNAs (miRNAs). Current research indicates that exosomes and miRNAs can serve as biomarkers and therapeutic targets for a variety of cardiovascular diseases (CVDs). This narrative review summarizes the research progress of exosomes and their miRNAs in CVDs, particularly in pulmonary valve diseases (PVDs), and, for the first time, explores their potential associations with transcatheter pulmonary valve replacement (TPVR). Currently, miRNAs play a crucial role in determining the optimal timing for TPVR intervention, and they demonstrate broad application prospects in post-TPVR right ventricular (RV) remodeling, treatment, and prognosis monitoring. However, the association between exosomes and miRNAs and the development of PVDs, particularly pulmonary regurgitation, remains unclear. The molecular mechanisms of exosomes and miRNAs in PVDs and RV remodeling after TPVR have not been fully elucidated, and their application in postoperative treatment following TPVR is still in its infancy. Future research must focus on advancing fundamental studies, validating biomarkers, and enhancing clinical applications to achieve significant breakthroughs.

## 1. Introduction

Cells secrete extracellular vesicles (EVs), which are membrane-bound structures containing various molecules [1,2]. The three major subtypes of EVs are exosomes, microvesicles, and apoptotic bodies, based on their biogenesis, biological characteristics, and receptor composition [3]. Studies indicate that exosomes have a diameter of 30 to 150 nm and possess high stability, low immunogenicity, high biocompatibility, and strong membrane penetration capability [4,5]. Exosomes are widely present in blood and lymph and can carry proteins, lipids, mRNA, lncRNA, and microRNAs (miRNAs) [6]. Therefore, they can regulate the pathological and physiological states of both local and distant cells [1,2].

Current research indicates that miRNA can serve as diagnostic biomarkers for a number of cardiovascular diseases (CVDs), such as aortic valve stenosis [7], pulmonary artery hypertension (PAH) [8], heart failure (HF) [9], and acute myocardial infarction [10]. Additionally, they can function as drug delivery vehicles, therapeutic agents, and drug targets [4,5,11]. Moreover, miRNAs contribute to the occurrence, development [12,13], and prognosis [14,15], as well as right ventricular (RV) remodeling of congenital heart disease (CHD) [16,17]. CHD affects approximately one-fifth of patients with right ventricular outflow tract (RVOT) obstruction [18]. Most patients experience pulmonary regurgitation (PR) several years after RVOT reconstruction [19]. Currently, transcatheter pulmonary valve replacement (TPVR), also known as percutaneous pulmonary valve implantation, is the primary treatment for patients with RVOT dysfunction and severe PR. As early as 2000, Bonhoeffer et al. completed the world’s first TPVR, successfully treating a child who had CHD and was experiencing valve dysfunction after surgery [20,21]. Studies demonstrate that the survival rates of patients undergoing TPVR are comparable to those of patients undergoing surgical pulmonary valve replacement [22]. However, compared to surgical pulmonary valve replacement, TPVR has the advantages of minimal invasion, rapid recovery, and high safety. This narrative review not only summarizes the research progress of exosomes and miRNAs in CVDs but also explores the potential relationship between them and pulmonary valve diseases (PVDs) and TPVR.

## 2. Physiological Characteristics of Exosomes

### 2.1. The Biogenesis of Exosomes

The biogenesis of exosomes begins with the invagination of the cell membrane, where the cell internalizes external substances or components from its surface through the process of endocytosis, resulting in early sorting endosome formation (Figure 1A) [23,24]. Ubiquitination plays a pivotal role in the formation of exosomes. It has been shown that proteins ubiquitinated in the Lys63 branching mode are preferentially transported to nascent exosomes [25,26]. This ubiquitination pattern typically does not lead to protein degradation by the proteasome but rather favors the loading of proteins into exosomes [25]. In contrast, proteins ubiquitinated in a Lys48 branching pattern are targeted for degradation [27]. The endosomal sorting complexes required for transporting complexes play a central role in the formation of exosomes by specifically recognizing ubiquitinated tags to sort cargo proteins [25,28]. This ensures that only correctly tagged proteins are loaded into exosomes. The membrane of the early endosome buds inward, forming intraluminal vesicles (Figure 1B) [29]. As the number of intraluminal vesicles increases, the early sorting endosome gradually transforms into a late sorting endosome, which subsequently evolves into a multivesicular body containing multiple vesicles (Figure 1B,C) [30]. During this process, the cell also recruits a variety of molecular components, such as membrane proteins and RNAs, into the vesicles (Figure 1A). The fusion of the multivesicular body with the cell membrane releases the intraluminal vesicles into the extracellular space, which in turn forms exosomes (Figure 1D). In general, exosomes range from 30 to 150 nm in diameter [31,32,33]. They facilitate communication between cells and material exchange, playing a crucial role both physiologically and pathologically [1].

### 2.2. The Molecular Composition of EVs

As important mediators of intercellular communication, exosomes possess complex and diverse contents that form the basis for their biological functional diversity. Exosomes contain a variety of proteins, including membrane proteins, cytoplasmic proteins, nuclear proteins, and extracellular matrix proteins (Figure 1D). They are also rich in various RNA components, including mRNA, miRNA, and lncRNA, and may even contain DNA fragments (Figure 1D) [34,35,36,37]. The surface markers of exosomes primarily include CD9, CD63, and CD81, which belong to the tetraspanin family of transmembrane proteins (Figure 1D) [29,38,39,40]. There is significant heterogeneity in their expression across different cell types and environments [29,38,39,40]. Recent studies have shown that circulating miRNAs and tissue miRNAs exhibit significant potential in the diagnosis, prognosis, and pathogenesis of CHD [41,42,43,44]. Furthermore, they contribute significantly to the occurrence and development of the tetralogy of Fallot (TOF) [45,46]. Additionally, when patients with CHD experience fluctuations in their condition or develop complications such as HF or other serious illnesses, the expression levels of specific circulating miRNAs can show significant changes [25,47,48]. These dynamic changes can not only assist clinicians in assessing disease progression, predicting treatment outcomes, and formulating personalized treatment but also enable better judgment regarding the timing of interventional treatments.

## 3. The Role of miRNA in CVDs

In recent years, significant advancements have been made regarding miRNAs in aortic valve, mitral valve, and tricuspid valve diseases. Aortic valve calcification is a prevalent valvular disease, impacting approximately 25% of the adult population [49]. Yanagawa et al. [50] discovered that the expression of miRNA-141 is decreased in patients with bicuspid aortic valve-related aortic stenosis, and it is associated with an increase in the level of bone morphogenetic protein-2, a key regulator of aortic valve calcification (Table 1). Zhang et al. [51] found that miRNA-30b can negatively regulate Runx2, Smad1, and caspase-3, thereby regulating aortic valve calcification (Table 1). Research has shown that miR-125b is significantly upregulated in aortic valves with severe stenosis compared with controls [52] (Table 1). Hosen et al. [53] identified a correlation between elevated levels of miRNA-122-5p and the absence of improvement in left ventricular function after transcatheter aortic valve replacement (TAVR) (Table 1). Additionally, this miRNA can be transported to cardiomyocytes via EVs and regulates cell apoptosis. Previous research has demonstrated a negative correlation between miRNA-206 levels and the left ventricular ejection fraction (LVEF) in patients after TAVR [54] (Table 1). It may thus be possible to assess risk in patients with calcifications of the aortic valve and predict cardiac function following TAVR by evaluating the expression levels of these miRNAs [55].

In terms of the mitral and tricuspid valves, Chen et al. [70] first discovered that the analysis of miRNA in valve tissue can distinguish between two typical degenerative mitral valve diseases, myxomatous mitral valve prolapse and fibroelastic deficiency. Chen et al. [56] found that the significant downregulation of miR-148b-3p and miR-409-3p is associated with the development of HF in patients with mitral regurgitation, providing a new perspective for the clinical prediction and intervention of HF (Table 1). In addition, miRNAs in exosomes can predict the recovery of cardiac function in patients with severe mitral regurgitation undergoing MitraClip repair [57,71] (Table 1). Hinojar et al. [58] found that, in comparison to the control group, the expression levels of miR-186-5p, miR-30e-5p, and miR-152-3p were significantly reduced in patients diagnosed with severe functional tricuspid regurgitation, suggesting their potential as diagnostic biomarkers for functional tricuspid regurgitation (Table 1).

Despite significant advancements in the exploration of miRNAs related to the aortic, mitral, and tricuspid valves, research focusing on miRNAs in PVD remains relatively scarce. Recent studies have indicated that miRNAs may play a regulatory role in the occurrence and development of pulmonary atresia [72,73]. Mirra et al. [74] found that hsa-miR-34a-5p may affect the function of the pulmonary valve by regulating cyclic nucleotide signal transduction pathways. De Carvalho et al. [75] found that the rs3746444 variation in the hsa-miR-499a/b gene results in alterations in the patient’s expression profile, which is a common phenotypic characteristic of RAS disease, congenital cardiomyopathy, and pulmonary artery stenosis. Huang et al. [76] proposed that key genes derived from EVs, identified through weighted gene co-expression network analysis, could serve as potential biomarkers for predicting congenital pulmonary stenosis. Zhang et al. [8] found that miR-8078 is upregulated in patients with CHD-PAH, suggesting that miR-8078 may be a potential therapeutic target or biomarker (Table 1). Therefore, miRNAs not only affect the function of the pulmonary valve by directly regulating the expression of genes related to the pulmonary valve but also may act indirectly on the pulmonary valve through their potential effects on RV function and involvement in cardiac development and signaling pathways.

TPVR is primarily applied to patients with RVOT dysfunction following RVOT reconstruction surgery, the most common of which is patients who have undergone correction for TOF. However, in recent years, the decrease in the number of children with RVOT dysfunction has led to a lower incidence of this condition, thereby limiting related basic and clinical research. Compared to the aortic valve and mitral valve, the related devices and technologies for TPVR are less mature, and the clinical trial products are relatively limited. Furthermore, research on exosomes and miRNAs in PVDs is still in the early phase. These factors collectively contribute to the relative scarcity of research on PVDs.

Future research should focus on the following directions. To begin with, it is imperative to conduct additional fundamental research to explore the roles of exosomes and miRNAs in PVDs and to elucidate the associated molecular mechanisms (Figure 2). Secondly, by conducting preclinical trials using animal models of PR, we can more accurately assess the therapeutic effects of exosomes and miRNAs in PVDs. Thirdly, it is necessary to explore the expression patterns of miRNA biomarkers before and after TPVR (Figure 2). Furthermore, to substantiate the biomarker potential of miRNAs, it is imperative that extensive clinical trials be undertaken through multicenter collaborations. In addition, we should integrate miRNAs with traditional cardiac function biomarkers, including troponin, BNP, and NT-proBNP, to comprehensively assess the prognosis of patients [77]. To enhance the validation and reproducibility of miRNA biomarkers, we plan to adopt the following measures in our future research. Firstly, we will expand the sample size to encompass a broader range of patient types. Secondly, we will employ multiple validation methods, including independent cohort validation and cross-validation. Lastly, we will actively seek collaboration with other research teams to jointly validate and share data. Through these measures, we hope to further enhance the reliability and practicality of miRNA biomarkers.

## 4. miRNA Aids in Determining the Optimal Timing for TPVR Intervention

Patients with RVOT dysfunction, such as TOF or transposition of the great arteries (TGA), may experience RV enlargement, increased volume overload, and PR several years after surgery [78]. As a result, the risk of right HF significantly increases. The timing of the intervention should consider both the protection of right heart function and the reduction in the intervention frequency. However, the optimal timing for performing TPVR is currently unclear.

According to the guidelines published by the European Society of Cardiology in 2020, for symptomatic patients with severe PR and/or at least moderate RVOT obstruction, the implementation of surgical pulmonary valve replacement or TPVR is recommended [79]. Research indicates that the circulating miRNA expression profiles in patients who have undergone surgery for TOF or TGA differ from those in healthy individuals [59,60,61,80,81] (Table 1). Liang et al. [45] demonstrated that the downregulation of miRNA-940 facilitates the proliferation and migration of secondary heart field progenitor cells (Table 1). This process may influence the occurrence and development of TOF, revealing its potential molecular mechanisms. Weldy et al. [47] found that TOF patients had RV enlargement and systolic dysfunction, and increased expression of miR-28-3p, miR-433-3p, and miR-371b-3p (Table 1). These features help monitor RV remodeling and disease progression in patients. Clouthier et al. [82] highlighted the potential of miRNAs as non-invasive biomarkers. They can reflect the systemic response to changes in right heart pressure in children with TOF and associated major aortopulmonary collaterals, providing important insights for tracking disease progression. Abu-Halima et al. [48] found that miRNA-183-3p can serve as an independent predictor of worsening HF in adult patients diagnosed with TGA and a systemic RV (Table 1). Lai et al. [60] found significant differences in the serum miRNA expression levels between patients who underwent atrial switch operation for complete TGA and the control population (Table 1). Notably, the serum levels of miR-18a and miR-486-5p were associated with systemic RV function. Other studies indicate that circulating miR-423-5p cannot be considered a biomarker for systemic ventricular function in adults following atrial repair for TGA [83]. In summary, specific miRNA expression profiles can reflect changes in right heart function in patients with CHD after surgery, particularly those with RVOT dysfunction. This helps determine the timing of TPVR. However, not all circulating miRNAs can serve as effective indicators for monitoring cardiac function, and their specific applications require further validation.

## 5. Exosomes, miRNAs, and Post-TPVR

### 5.1. The Impact of TPVR on RV Function

TOF is the most common cyanotic CHD, accounting for 20% of all CHDs [84,85]. In addition, CHDs such as complete transposition of the great arteries, double outlet right ventricle, and pulmonary valve stenosis, which involve abnormalities in the RVOT, also require reconstruction of the RVOT and repair of the pulmonary valve [86]. However, these patients often experience PR following reconstruction of RVOT, leading to increased preload on the RV, and RV hypertrophy, which can severely result in right HF [87,88]. TPVR can alleviate the afterload on the RV, reduce pulmonary artery pressure, and improve the long-term prognosis of patients.

Long-term follow-up indicates that the Melody and SAPIEN valves used for TPVR can effectively address RVOT dysfunction [89]. Additionally, patients with residual obstruction or infective endocarditis (IE) after the TPVR have a higher probability of re-valve replacement. Rużyłło et al. [90] conducted a follow-up study on 100 patients who underwent TPVR and found that only 14% of patients experienced serious adverse events during an average observation period of 5.5 years, with IE being the primary cause for reintervention. McElhinney et al. [91] conducted a large multicenter cohort study to assess the mid-term and long-term outcomes in 2476 patients who underwent TPVR. They found that the cumulative mortality rate 8 years after TPVR was 8.9%, with the primary cause of death being HF [91]. Additionally, the cumulative incidence of any transcatheter pulmonary valve reintervention by the eighth postoperative year was 25.1% [91]. The risk factors for surgical reintervention include age, previous IE, implanted stented bioprosthetic valves, and the postimplant gradient [91].

However, various complications following TPVR, including valve migration or embolism and coronary artery compression, may directly or indirectly damage myocardial tissue [92,93,94]. Compression of the coronary artery can lead to MI, which may then cause necrosis or apoptosis of myocardial cells, subsequently releasing multiple myocardial injury markers. Additionally, IE, as one of the serious complications following TPVR, can elicit an inflammatory response that is not limited to local myocardial necrosis. It may also exacerbate widespread myocardial injury through the activation of the immune system [95]. McElhinney et al. [96] conducted a follow-up study on 2476 patients from 15 centers between 2005 and 2020 and found that the incidence of IE after TPVR was 9.6% at 5 years and 20% at 10 years. Moreover, a history of previous endocarditis, younger age, and high residual pressure gradients were identified as important risk factors for IE. As research advances, innovative therapeutic approaches, including the utilization of exosomes and miRNAs, may be incorporated into TPVR surgery in the future, thereby providing novel strategies for enhancing RV function.

### 5.2. Role of Exosomes and miRNAs in RV Remodeling Following TPVR

Due to their complex anatomical structures, most CHDs often lead to RV myocyte hypertrophy, apoptosis, and fibrosis, ultimately potentially causing RV dysfunction [97]. Current research indicates that patients with severe PR after CHD corrective surgery experience improvements in RV function and favorable remodeling following TPVR [98,99]. Additionally, TPVR can alleviate RVOT obstruction and promote the functional recovery of the ventricle as well as the process of cardiac remodeling [93,100,101].

In recent years, the discovery of molecules such as miRNAs and exosomes has offered new insights into elucidating the pathological mechanisms of RV remodeling and developing intervention strategies. Shi et al. [62] found that mice subjected to hypoxia and pulmonary artery banding developed right HF, accompanied by downregulation of miR-223 expression in vivo (Table 1). Further research has shown that miR-223 inhibits the proliferation and migration of pulmonary vascular smooth muscle cells, reduces pulmonary vascular remodeling, and alleviates RV pressure and load by targeting IGF-1R and inhibiting its downstream signaling pathways [62]. This provides new therapeutic targets for the treatment of right HF caused by hypoxia and ischemia. Ma et al. [63] discovered that the overexpression of miR-335-5p can effectively promote the downregulation of calumenin, thereby accelerating the processes of cellular apoptosis and RV remodeling (Table 1). This finding opens up a new avenue for exploring therapeutic strategies for the right HF induced by PAH. Yuan et al. [64] found that miR-378 is secreted from cardiomyocytes after mechanical stress and inhibits the p38 MAPK-Smad2/3 signaling pathway by targeting MKK6, thereby slowing down or preventing the progression of fibrosis (Table 1). DiLorenzo et al. [17] discovered that MMP1 in exosomes derived from human amniotic epithelial cells can serve as a biomarker for RV remodeling after TOF repair, providing new tools for clinical monitoring and prognostic assessment. However, the role of miRNA in the process of RV remodeling after TPVR and the related molecular mechanisms remain unclear. In the future, it is essential to further elucidate the specific molecular mechanisms of these exosomes and miRNAs in RV remodeling following TPVR, as well as to develop targeted therapeutic strategies based on these molecules (Figure 2).

### 5.3. Application of Exosomes and mRNAs in Post-TPVR Monitoring

Studies have shown that the surface protein profile of plasma circulating EVs can be used as a non-invasive biomarker for heart transplantation rejection [102,103]. Although the research on exosomes in the field of organ transplantation is increasing, the research reports on transplant tissue-specific exosomes in the monitoring of immune rejection after TPVR are relatively scarce. Stoica et al. [67] found that the expression level of miR-1 increased in pediatric patients with CHD after surgery, and it was associated with cardiopulmonary bypass-related injury (Table 1). Therefore, miR-1 holds promise as a novel circulating biomarker. Zloto et al. [68] found that lower preoperative levels of miRNA-208a in pediatric patients with CHD were associated with a higher risk of postoperative complications (*p* = 0.03) (Table 1). Zloto et al. [69] also found that pediatric patients with neurologic deficits after heart surgery exhibited an increase in the level of miRNA-124a (Table 1). This suggests that miRNA-124a could serve as a potential biomarker.

Despite the promising application prospects of miRNAs in monitoring cardiac surgery, research on miRNAs in the field of TPVR is almost in its infancy (Figure 2). Furthermore, we do not yet know which specific miRNAs are closely associated with the success rate, postoperative complications, and long-term prognosis of TPVR (Figure 2).

### 5.4. Exosomes, miRNAs, and Post-TPVR Treatment

Exosomes, as an emerging acellular therapeutic strategy, have the potential to treat various CVDs [104,105,106,107]. Currently, exosomes as drug delivery carriers offer several advantages, including low immunogenicity, good safety, ease of storage and transportation, wide availability, multifunctionality, targeting capability, potential for mass production, and minimal ethical restrictions.

Beltrami et al. [65] found that exosomes in pericardial effusion promote cardiac angiogenesis by delivering miRNA-let-7b-5p to endothelial cells (Table 1). Research has shown that cardiac progenitor cells improve myocardial function through the paracrine secretion of exosomes, which contain numerous miRNAs that can promote angiogenesis and inhibit apoptosis [66] (Table 1). Barile et al. [108] discovered that the surface protein pregnancy-associated plasma protein A from exosomes derived from cardiac progenitor cells protects cardiomyocytes by mediating the release of insulin-like growth factor-1. In addition, microRNA-210 can not only inhibit apoptosis and promote angiogenesis but also improve the function of cardiomyocytes by controlling mitochondrial metabolism [109,110]. Therefore, microRNA-210 is considered a novel drug candidate for enhancing myocardial function. Bittle et al. [111] found that exosomes derived from human cardiosphere-derived cells can effectively protect right heart contractile function under conditions of overloading pressure. Gong et al. [112] found that exosomes derived from mesenchymal stem cells can promote cardiac repair after MI. Subsequently, they found that exosomes promote macrophage M2 polarization by targeting the miR-125a-5p/TRAF6/IRF5 signaling pathway, thereby facilitating the repair process. Additionally, Li et al. [113] were the first to develop a stem cell-derived exosome nebulization therapy for cardiac repair, confirming its efficacy in a mouse model of MI.

The drug delivery methods for exosomes include three strategies: exogenous drug loading, endogenous drug loading, and transgenic approaches [29,114,115]. Exogenous drug loading involves first isolating the exosomes and then introducing the drug into the exosomes through transfection or electroporation. Endogenous drug loading involves first introducing the drug into the cells. Once the drug enters the exosomes and is released from the donor cells, the drug-loaded exosomes are then isolated and purified. The transgenic approach involves genetically modifying the cells to express the desired drug, which is ultimately secreted outside the cells along with the exosomes. However, the application of exosomes in the treatment following transcatheter right heart valve repair and replacement is still in the early stages (Figure 2). In the future, it may play a significant role in promoting myocardial angiogenesis, inhibiting apoptosis, and protecting the myocardium, providing patients with safer and more effective treatment options.

The current clinical trials on EVs lack detailed reports on the techniques used, which affects the interpretation and reproducibility of the results [11,116]. Therefore, detailed reporting of research methods and in-depth investigation of EV subpopulations contribute to enhancing the translational success rate of EVs in clinical trials [116]. In the future, we need to establish efficient and sensitive detection technology platforms, such as high-throughput sequencing and mass spectrometry, for qualitative and quantitative analysis of exosomes and miRNAs. At the same time, we also need to develop relevant technical standards and operational norms to achieve standardization and normalization. Furthermore, before clinical application, it is indispensable to conduct a thorough assessment of the potential toxicity and immunogenicity of exosomes, and it is necessary to verify their therapeutic efficacy and safety through clinical trials.

## 6. Conclusions and Perspective

As important carriers of intercellular communication, exosomes play a significant role in the pathological mechanisms, biomarkers, and therapeutic efficacy assessment of aortic valve, mitral valve, and tricuspid valve diseases. The current review mainly focuses on the application of exosomes and miRNAs as biomarkers and therapeutic targets in the field of CVDs [117,118,119,120,121]. This narrative review not only summarizes the research progress of exosomes and their miRNAs in CVDs, particularly in PVDs, but also explores the potential relationship between them and PVD and TPVR for the first time. Currently, miRNAs play a crucial role in determining the optimal timing for TPVR intervention, and they demonstrate broad application prospects in post-TPVR RV remodeling, treatment, and prognosis monitoring. However, the specific miRNA biomarkers related to PVDs and RV function have not yet been identified, and the detailed molecular mechanisms of exosomes and their miRNAs in PVDs still await further elucidation. Additionally, the role of miRNAs in the success rate, postoperative complications, and long-term prognosis of TPVR also needs further exploration. Future research should focus on deepening fundamental studies, validating miRNA biomarkers, and promoting clinical application, with the aim of achieving breakthroughs in the field of PVDs.

## Figures and Tables

**Figure 1 ijms-25-13686-f001:**
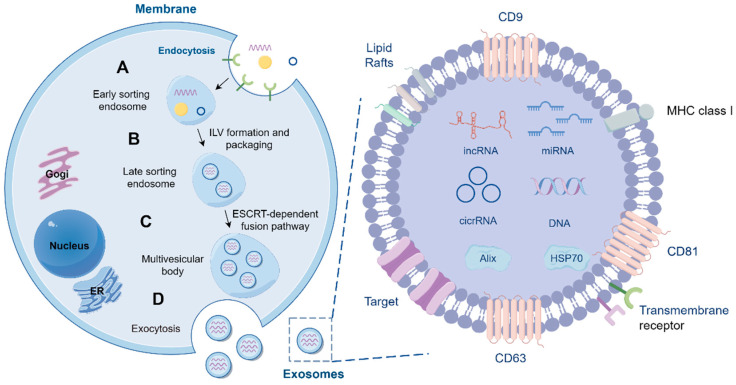
The diagram of the biogenesis of exosomes. (**A**) The formation of exosomes begins with the invagination of the cell membrane, through which the cell internalizes membrane proteins, RNA, and a variety of other molecules via endocytosis, resulting in the formation of early sorting endosomes. (**B**) After the membrane of early sorting endosomes buds inward to form intraluminal vesicles, it gradually transitions to a late sorting endosome. (**C**) Under the regulation of the endosomal sorting complexes required for transport complexes, the late sorting endosome transforms into a multivesicular body containing multiple vesicles. (**D**) The multivesicular body fuses with the cell membrane and releases intraluminal vesicles to the outside of the cell through exocytosis, forming exosomes. Exosomes contain a variety of protein and RNA components, and their surface markers mainly include CD9, CD63, and CD81.

**Figure 2 ijms-25-13686-f002:**
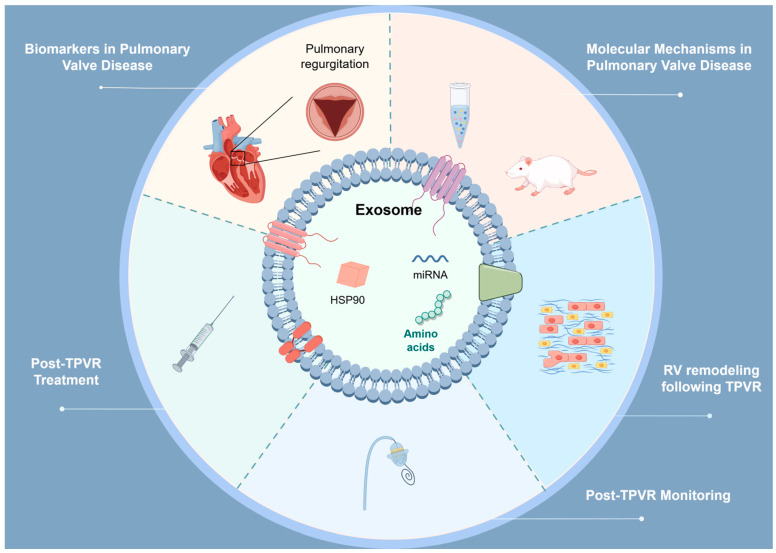
The gaps in the application of exosomes and miRNAs in pulmonary valve diseases (PVDs) and transcatheter pulmonary valve replacement (TPVR). Firstly, the association between exosomes and miRNAs and the development of PVDs, particularly pulmonary regurgitation (PR), remains unclear. It is currently unclear whether exosomes and miRNAs can serve as biomarkers for right ventricular (RV) function and PR. Secondly, the specific molecular mechanisms of exosomes and miRNAs in PVDs have not been fully elucidated. Thirdly, the specific molecular mechanisms of exosomes and miRNAs in RV remodeling following TPVR are also unclear. Fourthly, research on exosomes and mRNAs in post-TPVR monitoring is still in its early stages. Lastly, the application of exosomes and miRNAs in post-TPVR treatment has not yet matured.

**Table 1 ijms-25-13686-t001:** Summary of miRNA expression and its clinical significance in cardiovascular diseases.

miRNA	Sample Type	Expression Level	Clinical Significance	Reference
miRNA-141	aortic valve	Down	Decreased levels lead to a higher degree of aortic valve calcification	[50]
miRNA-30b	aortic valve	Down	Decreased levels lead to a higher degree of aortic valve calcification	[51]
miRNA-125b	aortic valve	Up	Increased levels lead to a higher degree of aortic valve calcification	[52]
miRNA-122-5p	blood	Up	Lack of improvement in left ventricular function after TAVR	[53]
miRNA-206	blood	Up	Decrease in LVEF after TAVR	[54]
miRNA-148b-3p, miRNA-409-3p	serum and left atrial tissue	Down	Biomarkers for heart failure in patients with mitral regurgitation	[56]
miRNA-21-5p	blood	Up	Increased levels lead to a higher degree of left ventricular reverse remodeling	[57]
miRNA-186-5p, miRNA-30e-5p, miRNA-152-3p	blood	Down	Diagnostic biomarkers for functional tricuspid regurgitation	[58]
miRNA-421, miRNA-1233-3p, miRNA-625-5p	blood	Down	Diagnostic biomarkers for TOF patients with symptomatic right HF	[59]
miRNA-18a, miRNA-486-5p	blood	Up	Decrease in systemic ventricular contractility for patients with complete TGA after surgery	[60]
miRNA-99b	blood	Up	Biomarkers for patients with repaired TOF and volume-overloaded RV	[61]
miRNA-766	blood	Down	Biomarkers for patients with repaired TOF and volume-overloaded RV	[61]
miR-8078	blood	Up	Biomarkers for patients with CHD-PAH	[8]
miRNA-940	RVOT tissue	Down	Decreased levels lead to the occurrence of TOF by targeting JARID2	[45]
miRNA-28-3p, miRNA-433-3p, miRNA-371b-3p	blood	Up	Biomarkers of increasing RV size and decreasing RV systolic function	[47]
miRNA-183-3p	blood	Up	Biomarkers of worsening HF in adult patients with TGA and a systemic RV.	[48]
miRNA-223	lung tissue from C57BL/6 mice	Down	Decreased levels lead to right HF by targeting IGF-IR	[62]
miRNA-335-5p	RV tissue	Up	Increased levels lead to RV remodeling by downregulating calumenin	[63]
miRNA-378	mice tissue	Up	Suppressed myocardial fibrosis by inhibiting the p38 MAPK-Smad2/3 signaling pathway	[64]
miRNA-let-7b-5p	blood, ascending aorta, right auricle	Up	Increased levels lead to angiogenesis	[65]
miRNA-210	right auricle tissue	Down	Inhibited apoptosis in cardiomyocytic cells	[66]
miRNA-132	right auricle tissue	Down	Enhanced tube formation in endothelial cells	[66]
miR-1	blood	Up	Biomarkers of cardiopulmonary bypass-related injury in pediatric patients with CHD after surgery	[67]
miRNA-208a	blood	Down	Biomarkers of high risk of postoperative complications in pediatric patients with CHD	[68]
miRNA-124a	blood	Up	Biomarkers of neurologic deficits in pediatric patients after heart surgery	[69]

CHD, congenital heart disease; HF, heart failure; LVEF, left ventricular ejection fraction; PAH, pulmonary arterial hypertension; RV, right ventricular; RVOT, right ventricular outflow tract; TAVR, transcatheter aortic valve replacement; TGA, transposition of the great arteries; TOF, tetralogy of Fallot.

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
