# Peer review of "Exosomes and miRNAs in Cardiovascular Diseases and Transcatheter Pulmonary Valve Replacement: Advancements, Gaps and Perspectives"

_ijms, 2024, doi:10.3390/ijms252413686_

Round 1

Reviewer 1 Report

Comments and Suggestions for Authors

This study reviews the role of exosomes. Exosomes are important mediators that transport microRNAs (miRNAs) and regulate physiological and pathological states of cells locally or remotely, but miRNA studies of pulmonary artery valve diseases (PVDs) have been scarce, and this report can contribute to future research and clinical practice.

On the other hand, I seek discussion on the following

1. The specific molecular mechanisms of how exosomes and miRNAs contribute to PVDs progression and RV remodeling seem insufficient for the present description. Please discuss possible molecular mechanisms.

2. There is a lack of validation and reproducibility of biomarkers for diagnosis and prognosis prediction using miRNA. Please point out future needs for validation of biomarkers.

3. Approaches for treatment and prognosis improvement after TPVR using exosomes and miRNAs are not well established. Please indicate what is needed for clinical application in the future.

4. Compared to other heart valve diseases, there seems to be less research specific to PVDs and less accumulation of disease-specific knowledge. Please discuss this point.

5. More in-depth recommendations are needed on bridging basic research on exosomes and miRNAs to facilitate their clinical application. Please discuss the linkage between basic and clinical research.

Reviewer 2 Report

Comments and Suggestions for Authors

This review article discusses associations of certain miRNAs with various cardiac valve diseases as well as CVDs, which are summarized in Table 1.  Besides superficially describing the biology of exosomes and miRNAs, the authors spend a large portion of the review on describing and discussing each of the miRNA in the table.  As such, there is very little substantial discussion on how exosomes are involved in pulmonary valve diseases except for their role as carriers of miRNAs.  Looking at the title and the abstract, however, the intended focus seems to be on the roles of (exosomes and) miRNAs in pulmonary valve diseases with some emphasis on TPVR-related topics.  As I went over the paper, it became so clear that not much is known on the role of miRNAs in pulmonary valve diseases (and the authors clearly state this in Abstract).  If there is not much infirmation available on a subject, wouldn’t you agree that it is too early to write a review focused on such an underdeveloped topic?  In several places, the authors comment on the need for more research.  Such comments are promotional statements, not reviews.  For these reasons, this reviewer feels rather negative toward this article.  That is to say a review focused on roles of miRNAs and exosomes in pulmonary valve diseases is premature as there is not much specific data to review and discuss.

Table 1 is a useful summary of changes in miRNAs associated with CVD.  It seems more reasonable to call this review miRNAs in CVD.  The authors should seriously consider changing the title and the focus statement so that this article is on a review of miRNAs on CVD or lung-related CVD.  There are several recently published reviews on miRNAs in CVD, and these reviews must be cited.  In doing so, the authors should state the uniqueness of their review compared with the recent reviews by others.   More specific comments are listed below.

1.       There are so many acronyms, which makes it difficult to read.  Use acronyms that are common, and avoid making new ones.  Spell out all the uncommon acronyms.  This will make reading so much easier.  Make a list of acronyms in an alphabetical order.

2.       The so-called graphical abstract does not represent the content of this review.  This figure has  too much emphasis on TPVR as the review is NOT on TPVR.   Remove this figure.

3.       As I stated above, the review is on CVD and miRNAs, not on exosomes.  However, all figures are on exosomes.  Figure 1 may be kept, but remove Figure 2.  This figure does not add anything to what is described in the text.  Rather than focusing on exosomes, it may help to illustrate how miRNAs are made in cells.

4.       Lines 40-41.  “Trams et al.[6] discovered these small membrane-bound vesicles in the supernatant of sheep red blood cells and termed them exosomes.”   This is an incorrect statement.  Ref. 6 has nothing to do with red blood cells.  Besides, what do you mean by “the supernatant of sheep red blood cells”?  Mature red blood cells are not supposed to generate exosomes.

5.       Table 1 lists miRNAs associated with CVD and cardiac functions.  Many of the miRNAs are presumably found in blood, but some may come from other body fluids or biopsies.  It is important where these miRNAs are found.  This could be done in the table (if there is enough space) or in the text when each of them is discussed.  “Up-regulation and “Down-regulation” in Table 1 can be simply “Up” and “Down”.

6.       Lines 171-172.  “if the intervention is performed too early, it may lead to a waste of resources.”  Indeed, this is a true statement.  However, as I see it, this is NOT the most important issue.  In medicine, the most important is to help patients.  So, the question to ask is, not economics, but whether or not an early intervention harms patients.  If it does, then it should be avoided.   If this helps them, then it is not such a big issue.  Please discuss the timing of intervention from the point of view of patient outcome.

7.       Section 6.1.  There is no discussion on miRNA nor exosomes.  What importance does this section have?

As Conclusion, the authors state, “However, the specific miRNA biomarkers related to PVDs and RV function have not yet been identified, and the detailed molecular mechanisms of exosomes and their miRNAs in PVDs still await further elucidation.”  If this is the conclusion of this review, the subject matter is NOT yet ready for being reviewed in any meaningful way.  One can certainly write an opinion piece or a commentary instead of a review.

Reviewer 3 Report

Comments and Suggestions for Authors

The authors present a comprehensive narrative review to summarize the research related to the role of exosomes and their miRNAs in the pathophysiology of pulmonary valve disease (PVD) and their potential associations with transcatheter pulmonary valve replacement (TPVR). The review is comprehensive and presents visually appealing graphics. Suggestions:

1) Abstract (line 18): please replace the word "study" with "narrative review" because this is not a research study

2) Table 1: Currently, the "clinical significance" column only lists the anatomical area affected, without explaining how the miRNA molecule is impacting that area. It is difficult to understand how each miRNA molecule listed in the first column is impacting the anatomical area listed in the "clinical significance" column. It would be helpful to expand the "clinical significance" column with more details about what is the clinical impact (e.g. increased levels lead to lower degree of valve calcification). 

3) Section 3 (line 111) is titled "The Role of miRNA in PVD". Yet, most of the content in this section (including Table 1) focusses on other anatomical targets in the cardiovascular system. Therefore, please consider relabeling this section to "The Role of miRNA in cardiovascular diseases" for accuracy.

4) Between lines 146-155, please elaborate the potential indirect effect of miRNA on the pulmonary valve (PV) through its potential impact on the RV; and the potential role of miRNA in PV atresia (congenital heart disease), PV insufficiency (hsa-miR-34a-5p > Cyclic nucleotides signal transduction), and PV stenosis

ref: Nappi F, Iervolino A, Avtaar Singh SS, Chello M. MicroRNAs in Valvular Heart Diseases: Biological Regulators, Prognostic Markers and Therapeutical Targets. Int J Mol Sci. 2021 Nov 9;22(22):12132. doi: 10.3390/ijms222212132. PMID: 34830016; PMCID: PMC8618095.

Mirra D, Cione E, Spaziano G, Esposito R, Sorgenti M, Granato E, Cerqua I, Muraca L, Iovino P, Gallelli L, D'Agostino B. Circulating MicroRNAs Expression Profile in Lung Inflammation: A Preliminary Study. J Clin Med. 2022 Sep 16;11(18):5446. doi: 10.3390/jcm11185446. PMID: 36143090; PMCID: PMC9500709.

de Carvalho JB, de Morais GL, Vieira TCDS, Rabelo NC, Llerena JC Jr, Gonzalez SMC, de Vasconcelos ATR. miRNA Genetic Variants Alter Their Secondary Structure and Expression in Patients With RASopathies Syndromes. Front Genet. 2019 Nov 13;10:1144. doi: 10.3389/fgene.2019.01144. PMID: 31798637; PMCID: PMC6863982.

5) In section 6 (TPVR), please comment on Transplant tissue specfic exosome platform for noninvasive monitoring of immunologic rejection [mRNA] in TPVR

6) In section 6, please also mention about miRNA's potential utility in prognostication and monitoring post-operatively, as an area of future research.

ref: Stoica SC, Dorobantu DM, Vardeu A, Biglino G, Ford KL, Bruno DV, Zakkar M, Mumford A, Angelini GD, Caputo M, Emanueli C. MicroRNAs as potential biomarkers in congenital heart surgery. J Thorac Cardiovasc Surg. 2020 Apr;159(4):1532-1540.e7. doi: 10.1016/j.jtcvs.2019.03.062. Epub 2019 Apr 4. PMID: 31043318.

Thank you for your submission.

Round 2

Reviewer 1 Report

Comments and Suggestions for Authors

Regarding this revised version, the authors have made appropriate corrections and additions to my previous comments. The corrections are appropriate. I have no further comments to make. I hope that this will be made public as soon as possible, as it will provide useful information to readers.

Reviewer 2 Report

Comments and Suggestions for Authors

This is a much-improved manuscript as it is focused and logically organized.  I have only a few suggestions.

1.       Please consider the following title:  Exosomes and miRNAs in Cardiovascular Diseases and Transcatheter Pulmonary Valve Replacement: Advancements, Gaps and Perspectives.

2.       There are many ways to present a list of abbreviations.  The way it is done by the authors is very difficult to find each entry.  Consider using the BOLD font for entries.  If certain abbreviations are used three times or less, do not abbreviate.

3.       Line 48.  “Cells secrete extracellular vesicles (EVs) that contain membrane structure.”  This is a strange statement.  In cell biology, vesicles are, by definition, membrane bound.  Do you mean that EVs contain membrane fragments? Or lipids?  As an introductory sentence, this is not an attractive one.

4.       Figure 1 and biogenesis of exosomes.  Readers may wonder if some contents in the endocytic vesicles are eventually in exosomes, how is this achieved?  The simple explanation of internal budding cannot explain how some materials in endocytic vesicle are in exosomes.  This is an important point in exosome biogenesis.  Please add this discussion to the text.

5.       Line 285.  5.1

6.       Line 386.  Remove one of the “of’s”.
